# Polar Ducks and Where to Find Them:
# Enhancing Entity Linking with Duck Typing and Polar Box Embeddings

**Mattia Atzeni**[1,2], **Mikhail Plekhanov**[1], **Frédéric A. Dreyer**[1], **Nora Kassner**[1],
**Simone Merello**[1], **Louis Martin**[1], **Nicola Cancedda**[1*]
[1]Meta AI, [2]EPFL

## Abstract

Entity linking methods based on dense re-
trieval are widely adopted in large-scale ap-
plications for their efficiency, but they can fall
short of generative models, as they are sensi-
tive to the structure of the embedding space.
To address this issue, this paper introduces
DUCK, an approach to infusing structural in-
formation in the space of entity representa-
tions, using prior knowledge of entity types.
Inspired by *duck typing* in programming lan-
guages, we define the type of an entity based
on its relations with other entities in a knowl-
edge graph. Then, porting the concept of box
embeddings to spherical polar coordinates, we
represent relations as boxes on the hypersphere.
We optimize the model to place entities in-
side the boxes corresponding to their relations,
thereby clustering together entities of similar
type. Our experiments show that our method
sets new state-of-the-art results on standard
entity-disambiguation benchmarks. It improves
the performance of the model by up to 7.9
$F_1$ points, outperforms other type-aware ap-
proaches, and matches the results of generative
models with 18 times more parameters.

## 1 Introduction

State-of-the-art approaches to entity linking,
namely the task of linking mentions of entities in
a text to the corresponding entries in a knowledge
base (KB) (Ferragina and Scaiella, 2010; Ganea
and Hofmann, 2017), are nowadays large *genera-*
*tive models* (Aghajanyan et al., 2022; De Cao et al.,
2021) which perform entity retrieval in a autore-
gressive way. This category of methods achieves
the best results, as it allows effectively capturing
relations between the context of a mention and en-
tity descriptions. However, the preferred choice in
large-scale applications are often methods based

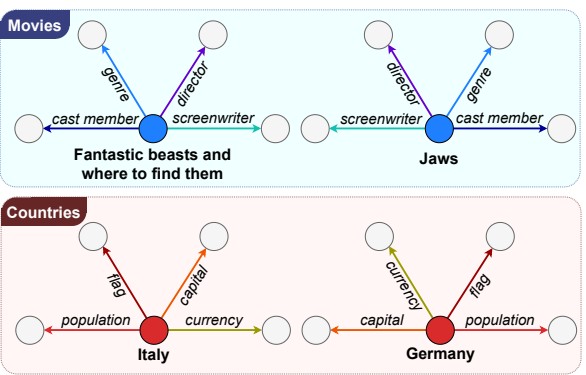

Figure 1: Examples from Wikidata showing how, fol-
lowing the concept of *duck typing*, relations in a knowl-
edge graph can help identifying entities of different
types (e.g., movies and countries).

on *dense retrieval* (Plekhanov et al., 2023; Botha
et al., 2020; Ayoola et al., 2022), as they are easier
to train and can be more than one order of magni-
tude faster (Ayoola et al., 2022). These approaches
learn to represent entities and mentions separately
in the same embedding space, so that, at inference
time, the method only requires encoding the men-
tion and retrieving the most similar entity. Methods
based on dense retrieval have the drawback of be-
ing very sensitive to the structure of the embedding
space, thereby reaching lower accuracy compared
to generative models (Aghajanyan et al., 2022).

In this paper, we aim to close the gap with gener-
ative approaches by infusing structural information
in the latent space of retrieval-based methods. Re-
cent work (Mulang et al., 2020; Ayoola et al., 2022)
has shown the benefit of infusing prior factual
knowledge in the models. In particular, Raiman
and Raiman (2018) reported that prior knowledge
of the type of a mention would result in nearly per-
fect disambiguation performance. Prior methods
used type labels extracted from knowledge graphs
(KGs) (Ayoola et al., 2022; Chen et al., 2020; Orr
et al., 2021), but as KGs can be highly incomplete,
we aim to define type information in a fuzzy and
more fine-grained manner.

---

[*]Correspondence to mattia.atzeni@epfl.ch and
ncan@meta.com. Work done while the first author was an
intern at Meta AI.

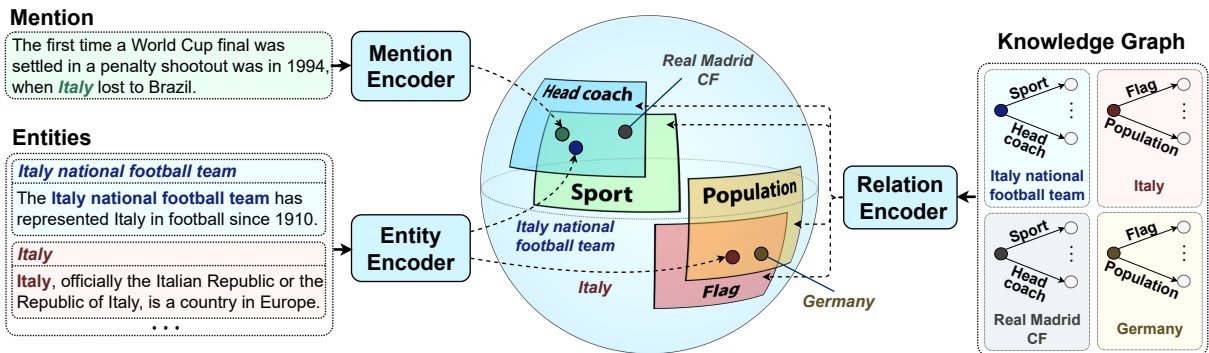

Figure 2: Entity disambiguation flow in DUCK. A mention encoder and an entity encoder learn to represent mentions and entity descriptions respectively. Following the concept of *duck typing*, relations in a knowledge graph are used to determine entity types. Relations are represented as box embeddings in spherical polar coordinates and the model is optimized to place entities inside the boxes corresponding to their relations.

We achieve this goal by drawing inspiration from the concept of *duck typing* in programming languages, which relies on the idea of defining the type of an object based on its properties. Extending this idea to the realm of KGs, we define the type of an entity based on the relations that it has with other entities in the graph. Figure 1 shows some examples demonstrating how relational information from a KG like Wikidata (Vrandečić and Krötzsch, 2014) can help identifying entities of different types without any need for type labels.

Motivated by this intuition, we propose DUCK (*Disambiguating Using Categories extracted from Knowledge*), an approach to infusing prior type information in the latent space of methods based on dense retrieval. Building on recent work on region-based representations (Dasgupta et al., 2020; Abboud et al., 2020), we introduce box embeddings in spherical polar coordinates and employ them to model relational information, as shown in Figure 2. We define this representation as it naturally aligns with the use of the dot product (or the cosine similarity) as the similarity function for entity and mention embeddings (see Section 3), which is the prevalent choice in dense-retrieval methods. Then, we optimize the model to structure the latent space in such a way that entities fall within the boxes corresponding to their relations, so that entities that share many relations (which are assumed to be of the same type) will be clustered together.

We use our approach to train a bi-encoder model with the same architecture as Wu et al. (2020). Our experiments show that DUCK sets new state-of-the-art results on standard entity-disambiguation datasets, exceeds the performance of other type-aware models (trained on 10 times more data), and

matches the overall results of more computationally intensive generative models, with 18 times more parameters than DUCK. Our ablation studies show that incorporating type information using box embeddings in polar coordinates improves the performance by up to 7.9 micro-$F_1$ points with respect to the same model trained without *duck typing*. Finally, qualitative analyses support the intuition that our method results in a clear clustering of types and that DUCK is able to predict the relations of an entity despite the incompleteness of the KG.

## 2 Preliminaries

We start by formalizing the entity-disambiguation problem, then we outline the main intuitions underlying methods based dense-retrieval.

**Problem statement.** The goal of *entity disambiguation* (ED) is to link entity mentions in a piece of text to the entity they refer to in a reference KB. For each entity $e$, we assume we have an entity description expressed as a sequence of tokens $s_e = (s_e^{(1)}, \ldots, s_e^{(|s_e|)})$. Similarly, each mention $m$ is associated with a sequence of tokens $s_m = (s_m^{(1)}, \ldots, s_m^{(|m|)})$, representing the mention itself and its context. We denote the entity a mention $m$ refers to as $e_m^\star$. Further, we assume that the reference KB is a knowledge graph $\mathcal{G} = (\mathcal{E}, \mathcal{R})$, where $\mathcal{E}$ is a set of entities and $\mathcal{R}$ is a set of relations, namely boolean functions $r : \mathcal{E} \times \mathcal{E} \to \{0, 1\}$ denoting whether a relation exists between two entities. Then, given a set of entity-mention pairs $\mathcal{D} = \{(m_1, e_{m_1}^\star), \ldots, (m_{|\mathcal{D}|}, e_{m_{|\mathcal{D}|}}^\star)\}$, we aim to learn a model $f : \mathcal{M} \to \mathcal{E}$, such that the entity predicted by the model for a given mention $\hat{e}_m = f(m)$ is the correct entity $e_m^\star$.

**Dense-retrieval methods.** Methods based on dense retrieval (Wu et al., 2020) learn to represent mentions and entities in the same latent space, often optimizing a cross-entropy loss of the form:

$$\mathcal{L}_{ED}(m) = -s(m, e_m^\star) + \log \sum_j \exp(s(m, e_j)),$$

where $s$ is a similarity function between entities and mentions. This objective encourages the representation of mention $m$ to be close to the representation of the correct entity $e_m^\star$ and far from other entities $e_j$, according to the similarity $s$. This similarity function $s(m, e)$ is usually chosen to be the dot product between learned representations $\boldsymbol{m}, \boldsymbol{e} \in \mathbb{R}^d$ of the mention and entity respectively. At inference time, a mention is encoded in the dense space of entity embeddings and the entity with the highest similarity is returned.

## 3 DUCK: enhancing entity disambiguation with duck typing

Our approach builds on dense-retrieval methods and aims to enhance their performance using fine-grained type information.

### 3.1 Modeling fine-grained type information from knowledge graphs

**Duck typing on knowledge graphs.** *Duck typing* is a well-known concept in dynamically typed programming languages and is based on the overall idea of weakly defining the type of an object based on its properties. Extending this concept to KGs, without any need for type labels, we can describe the type of an entity $e \in \mathcal{E}$ in terms of the set of relations labeling the edges originating from $e$ in the KG. With slight abuse of notation, we will denote this set as $\mathcal{R}(e) = \{r \in \mathcal{R} \mid \exists e' \in \mathcal{E} : r(e, e') = 1\}$. An example of how the set of relations of an entity can be used to determine its type is shown in Figure 2. For a qualitative analysis showing how duck typing works in real-world knowledge graphs, we refer the reader to Appendix A.

**Relations as polar box embeddings.** Inspired by region-based representations (Vendrov et al., 2016; Lai and Hockenmaier, 2017), and particularly by box embeddings (Vilnis et al., 2018; Li et al., 2019; Dasgupta et al., 2020), we represent relations as regions of the space. Our similarity function $s$, the dot product, is the product of the two norms of the entity and mention embeddings and the cosine of the angle between them. Since we are using this similarity to rank entities for a given mention, the norm of the mention embedding is irrelevant, whereas the entity norms encode a "prior" over entities. Therefore, we choose to represent relations as boxes in spherical polar coordinates, as shown in Figure 2. This representation allows guaranteeing that the cosine of the angle between two embeddings falling in the same region is constrained by the boundaries of the box. At the same time, it keeps boxes open on the radial coordinate, so as to leave the training free to use entity norms to encode prior probabilities without interference from type information. Concretely, we parameterize the box corresponding to a relation as a pair of vectors:

$$Box(r) = (\boldsymbol{\varphi}_r^-, \boldsymbol{\varphi}_r^+),$$

where $\boldsymbol{\varphi}_r^-, \boldsymbol{\varphi}_r^+ \in \mathbb{R}^{d-1}$ are vector of angles denoting respectively the bottom-left and top-right corners of the box in spherical coordinates. For an entity $e \in \mathcal{E}$, we say that $e \in Box(r)$, if the expression in polar coordinates $\boldsymbol{\varphi}_e$ of the entity representation $\boldsymbol{e}$ is between $\boldsymbol{\varphi}_r^-$ and $\boldsymbol{\varphi}_r^+$ across all dimensions. Then, our goal is to structure the latent space in such a way that $e \in Box(r^+)$ for every $r^+ \in \mathcal{R}(e)$ and $e \notin Box(r^-)$ for every $r^- \in \mathcal{R} \setminus \mathcal{R}(e)$.

### 3.2 Duck typing as an optimization problem

In order to achieve the goal mentioned above, we need to turn the intuition of Section 3.1 into an optimization problem. To this end, it helps to define a distance function between an entity and a box.

**Entity-box distance.** Following Abboud et al. (2020), who defined a similar function for box embeddings in cartesian coordinates, we define the distance between an entity and a box as:

$$dist(e, r) = \begin{cases} \|(\boldsymbol{\varphi}_e - \bar{\boldsymbol{\varphi}}_r)/(\boldsymbol{\delta}_r + 1)\|_2 & \text{if } e \in Box(r) \\ \|(|\boldsymbol{\varphi}_e - \bar{\boldsymbol{\varphi}}_r| \circ (\boldsymbol{\delta}_r + 1) - \boldsymbol{\kappa})\|_2 & \text{otherwise,} \end{cases}$$

where $\bar{\boldsymbol{\varphi}}_r = (\boldsymbol{\varphi}_r^- + \boldsymbol{\varphi}_r^+)/2$ is the center of the box corresponding to relation $r$, $\boldsymbol{\delta}_r = \boldsymbol{\varphi}_r^+ - \boldsymbol{\varphi}_r^-$ is a vector containing the width of the box along each dimension, $\circ$ is the Hadamard product, $/$ is element-wise division, and $\boldsymbol{\kappa}$ is a vector of width-dependent scaling coefficients defined as:

$$\boldsymbol{\kappa} = \frac{\boldsymbol{\delta}_r}{2} \circ (\boldsymbol{\delta}_r - \frac{1}{\boldsymbol{\delta}_r + 1} + 1).$$

Intuitively, this function heavily penalizes entities outside the box, with higher distance values and gradients, whereas it mildly pushes entities lying already inside the box towards the center. We refer the reader to Appendix B for more details.

**Loss function for typing.** To encourage an entity $e \in \mathcal{E}$ to lie inside all boxes representing the relations $\mathcal{R}(e)$ and outside the other boxes, we use a negative-sampling loss similar to the one of Sun et al. (2019). Our loss function is defined as:

$$\mathcal{L}_{Duck}(e) = -\mathbb{E}_{r^+}[\log \sigma(\gamma - dist(e, r^+))] \\ - \mathbb{E}_{r^-}[\log \sigma(dist(e, r^-) - \gamma)].$$

Above, $\gamma \in \mathbb{R}$ is a margin parameter, $\sigma$ is the sigmoid function, $r^+$ is a relation of entity $e$, drawn uniformly from the set of relations $\mathcal{R}(e)$, whereas $r^-$ is a relation drawn from the set of relations $\mathcal{R} \setminus \mathcal{R}(e)$ according to the probability distribution:

$$\hat{p}(r_i^- \mid e) = \frac{\exp(-\alpha \cdot dist(e, r_i^-))}{\sum_{r_j^- \in \mathcal{R} \setminus \mathcal{R}(e)} \exp(-\alpha \cdot dist(e, r_j^-))}$$

where $\alpha \in [0, 1]$ is a temperature parameter. The lower $\alpha$, the closer the distribution is to a uniform distribution, whereas higher values of $\alpha$ result in more weight given to boxes that are close to the entity. Notice that this objective forces the distance between an entity $e$ and relations $r^+ \in \mathcal{R}(e)$ to be small, while keeping the entity far from boxes corresponding to the negative relations $r^-$. Hence, optimizing the objective $\mathcal{L}_{Duck}$ will result in clustering together entities that share many relations.

**Overall optimization objective.** We train the model to optimize jointly the entity-disambiguation loss of Section 2 and the duck-typing loss $\mathcal{L}_{Duck}$. Although we defined the loss $\mathcal{L}_{Duck}$ for entities, we calculate it for mentions as well, defining the set of relations of a mention based on the ground-truth entity $\mathcal{R}(m) = \mathcal{R}(e_m^\star)$. In order to prevent boxes from growing too large during training, we further introduce an L2 regularization term $l_2$ on the size of the boxes:

$$l_2 = \frac{1}{d-1}\mathbb{E}_r[\boldsymbol{\delta}_r^\top \boldsymbol{\delta}_r].$$

Then, our final optimization objective is:

$$\mathcal{L}(m) = \mathcal{L}_{ED}(m) + \lambda_{Duck}(\mathcal{L}_{Duck}(e_m^\star) + \mathcal{L}_{Duck}(m) + \lambda_{l_2} l_2),$$

where $\lambda_{Duck}, \lambda_{l2} \in [0, 1]$ are hyperparameters defining the weight of each component of the loss.

## 4 A bi-encoder model with duck typing

Building on prior work, we used the method described in Section 3 to train a bi-encoder model with the same architecture of Wu et al. (2020). Compared to Wu et al. (2020), DUCK adds just a relation encoder which is only used at training time to represent relations as boxes.

### 4.1 Bi-encoder

Bi-encoders, introduced in this context by Wu et al. (2020), are an efficient architecture for approaches based on dense retrieval. These methods rely on two different encoders $f_{entity}$ and $f_{mention}$ to represent entities and mentions respectively.

**Entity encoder.** Given a textual description of an entity $e \in \mathcal{E}$, expressed as a sequence of tokens $s_e = (s_e^{(1)}, \ldots, s_e^{(|s_e|)})$, we learn an entity representation $\boldsymbol{e} \in \mathbb{R}^d$ as:

$$\boldsymbol{e} = f_{entity}(s_e).$$

Concretely, following prior work (Wu et al., 2020), we extract entity descriptions $s_e$ from Wikipedia, and we structure each description $s_e$ using the title of the Wikipedia page associated with entity $e$ followed by the initial sentences of the body of the page, separated by a reserved token. We truncate entity descriptions $s_e$ to a maximum sequence length of $n_e$. For the entity encoder $f_{entity}$, we used a pre-trained RoBERTa model (Liu et al., 2019), resorting to the encoding of the [CLS] token for the final entity representation $\boldsymbol{e}$.

**Mention encoder.** We model a mention as a sequence of tokens $s_m = (s_m^{(1)}, \ldots, s_m^{(|s_m|)})$ denoting both the mention itself and the context surrounding it, up to a maximum mention length $n_m$. Following Wu et al. (2020), we used reserved tokens to denote the start and the end of a mention and separate it from the left and right context. We then calculate mention representations as:

$$\boldsymbol{m} = f_{mention}(s_m),$$

where $f_{mention}$ is a mention encoder based on a pre-trained RoBERTa model and the final mention representation $\boldsymbol{m}$ is obtained using the encoding of the [CLS] token. Overall, our bi-encoder is the same as the one used by Wu et al. (2020), with the only difference that we rely on RoBERTa instead of BERT (Devlin et al., 2019) as the underlying language model.

### 4.2 Relation encoder

**Relation modeling.** We model a relation $r \in \mathcal{R}$ as a sequence of tokens $s_r = (s_r^{(1)}, \ldots, s_r^{(|s_r|)})$. These sequences are extracted from Wikidata (Vrandečić and Krötzsch, 2014), using the English label of the property and its description, separated by a reserved token. We used the same mapping from Wikipedia titles to Wikidata identifiers of De Cao

et al. (2021). Based on $s_r$, we then compute a relation embedding $r$ for each relation $r \in \mathcal{R}$ as:

$$r = f_{relation}(s_r),$$

where $f_{relation}$ is a relation encoder similar to $f_{entity}$ and $f_{mention}$, which computes the relation representation $r$ as the embedding of the `[CLS]` token produced by a pre-trained RoBERTa model.

**Learning boxes in polar coordinates.** Given a relation representation $r$ calculated as described above, we parametrize a box as a pair of vectors $Box(r) = (\boldsymbol{\varphi}_r^-, \boldsymbol{\varphi}_r^+)$, where:

$$\boldsymbol{\varphi}_r^- = \sigma(\text{FFN}^-(r)) \cdot \pi$$
$$\boldsymbol{\varphi}_r^+ = \boldsymbol{\varphi}_r^- + \delta_{\min} + \sigma(\text{FFN}^+(r)) \cdot (\pi - \boldsymbol{\varphi}_r^- - \delta_{\min}).$$

Above, $\text{FFN}^-$ and $\text{FFN}^+$ are 2-layer feed-forward networks, $\sigma$ is the sigmoid function, and $\delta_{\min}$ is a margin parameter denoting the minimum width of a box across any dimension. Calculating the corners of a box in this manner allows us to achieve two main objectives: *(i)* all components of $\boldsymbol{\varphi}_r^-$ and $\boldsymbol{\varphi}_r^+$ range from 0 to $\pi$, hence they assume valid values in the spherical coordinate system, and *(ii)* $\boldsymbol{\varphi}_r^+$ is greater than $\boldsymbol{\varphi}_r^-$ across all dimensions, so that boxes are never empty and the model does not have to learn how to produce non-degenerate regions. Notice that, in a spherical coordinate system, only one of the coordinates is allowed to range from 0 to $2\pi$, while all remaining coordinates will range from 0 to $\pi$. For simplicity, we constrain all coordinates in the interval $[0, \pi]$, thereby reducing all representations to half of the hypersphere.

### 4.3 Training and inference

**Training.** We train DUCK by optimizing the overall objective defined in Section 3. In order to compute the loss $\mathcal{L}_{Duck}$, we calculate the representations $\boldsymbol{\varphi}_e, \boldsymbol{\varphi}_m \in \mathbb{R}^{d-1}$ by converting to spherical coordinates the entity and mention representations $e$ and $m$ produced by the entity and mention encoders respectively. To make training more efficient, the relation representations $r$ are pre-computed and kept fixed at training time. We use the dot product between entity and mention representations to evaluate the entity disambiguation loss $\mathcal{L}_{ED}$:

$$s(e, m) = e^\top m.$$

The expectations in the loss $\mathcal{L}_{Duck}$ are estimated across all relations $r^+ \in \mathcal{R}(e)$ and by sampling $k$ relations $r^- \in \mathcal{R} \setminus \mathcal{R}(e)$ according to $\hat{p}(r^- \mid e)$. The L2 regularization on the width of the boxes is performed across all relations in a batch.

**Inference.** At inference time, our approach is not different from the method of Wu et al. (2020). We simply match a mention $m$ to the entity that maximizes the similarity function $s$:

$$\hat{e}_m = \arg\max_{e \in \mathcal{E}_m} s(e, m),$$

where $\mathcal{E}_m \subseteq \mathcal{E}$ is a set of candidate entities for mention $m$. In practice, we can precompute all entity embeddings, so that inference only requires one forward pass through the mention encoder and selecting the entity with the highest similarity.

## 5 Experiments

This section provides a thorough evaluation of our approach. First, we show that DUCK achieves new state-of-the-art results on popular datasets for entity disambiguation, closing the gap between retrieval-based methods and more expensive generative models. Then, we discuss several ablation studies, showing that incorporating type information using box embeddings in polar coordinates improves the performance of the model. Finally, we dig into qualitative analyses, showing that our model is able to place entities in the correct boxes despite the incompleteness of the information in the KG.

### 5.1 Experimental setup

We reproduce the same experimental setup of prior work (De Cao et al., 2021; Le and Titov, 2019): using the same datasets, the same candidate sets, and comparing the models based on the *InKB* micro-$F_1$ score. Following De Cao et al. (2021); Wu et al. (2020), we train the model on the BLINK data (Wu et al., 2020), consisting of 9M mention-entity pairs extracted from Wikipedia. Entity descriptions are taken from the Wikipedia snapshot of Petroni et al. (2021). Then, we measure *in-domain* and *out-of-domain* generalization by fine-tuning the model on the training set of the AIDA-CoNLL dataset and evaluating on six test sets: **AIDA** (Hoffart et al., 2011), **MSNBC** (Cucerzan, 2007), **AQUAINT** (Milne and Witten, 2008), **ACE2004** (Ratinov et al., 2011), **CWEB** (Gabrilovich et al., 2013) and **WIKI** (Guo and Barbosa, 2018).

### 5.2 Entity disambiguation results

We compared DUCK against three main categories of approaches: *(a)* methods based on *dense retrieval*, *(b) generative models*, and *(c) type-aware models*, namely other approaches to adding type

| Method | AIDA | MSNBC | AQUAINT | ACE2004 | CWEB | WIKI | Avg. |
|---|---|---|---|---|---|---|---|
| *Dense retrieval* | | | | | | | |
| Ganea and Hofmann, 2017 | 92.2 | 93.7 | 88.5 | 88.5 | 77.9 | 77.5 | 86.4 |
| Yang et al., 2018 | **95.9** | 92.6 | 89.9 | 88.5 | **81.8** | 79.2 | 88.0 |
| Shahbazi et al., 2019 | 93.5 | 92.3 | 90.1 | 88.7 | 78.4 | 79.8 | 87.1 |
| Yang et al., 2019 | 93.7 | 93.8 | 88.2 | 90.1 | 75.6 | 78.8 | 86.7 |
| Le and Titov, 2019 | 89.6 | 92.2 | 90.7 | 88.1 | 78.2 | 81.7 | 86.8 |
| Fang et al., 2019 | 94.3 | 92.8 | 87.5 | 91.2 | 78.5 | 82.8 | 87.9 |
| Wu et al., 2020[†] | 79.6 | 80.0 | 80.3 | 82.5 | 64.2 | 75.5 | 77.0 |
| *Generative models* | | | | | | | |
| De Cao et al., 2021 | 93.3 | 94.3 | 89.9 | 90.1 | 77.3 | 87.4 | 88.8 |
| Aghajanyan et al., 2022 (*CM3-Medium*) | 93.5 | 94.2 | 90.1 | 90.4 | 76.5 | 86.9 | 88.6 |
| Aghajanyan et al., 2022 (*CM3-Large*) | 94.8 | 94.8 | 91.1 | 91.4 | 78.4 | **88.7** | 89.8 |
| *Type-aware models* | | | | | | | |
| Chen et al., 2020 | 93.7 | 94.5 | 89.1 | 90.8 | 78.2 | 81.0 | 86.7 |
| Orr et al., 2021[‡] | 80.9 | 80.5 | 74.2 | 83.6 | 70.2 | 76.2 | 77.6 |
| Ayoola et al., 2022 (*Wikipedia*) | 87.5 | 94.4 | **91.8** | 91.6 | 77.8 | **88.7** | 88.6 |
| Ayoola et al., 2022 (*fine-tuned*) | 93.9 | 94.1 | 90.8 | 90.8 | 79.4 | 87.4 | 89.4 |
| DUCK (*Wikipedia*) | 91.0 | **95.1** | 91.3 | **95.4** | 76.9 | 86.1 | 89.3 |
| DUCK (*fine-tuned*) | 93.7 | 94.6 | 91.3 | 95.0 | 78.2 | 85.9 | **89.8** |

Table 1: Micro-$F_1$ (*InKB*) results on six entity-disambiguation datasets. **Bold** indicates the best model, underline indicates the second best results. Our results are highlighted in gray. [†]Model without candidate set, results from De Cao et al. (2021). [‡]Results from Ayoola et al. (2022).

information to retrieval-based methods (DUCK pertains to this category). We report the results both for the model trained only on the BLINK data and for the model fine-tuned on AIDA, referring to the former as "DUCK (*Wikipedia*)" and to the latter as "DUCK (*fine-tuned*)".

**Main results.** Table 1 shows the performance of DUCK in comparison to other methods. First, we notice that DUCK obtains state-of-the-art results on **MSNBC** and **ACE2004**, second best performance on **AQUAINT**, and state-of-the-art results on average across all datasets. We also observe that DUCK outperforms all the other type-aware models, showing the effectiveness of our approach to define type information and infuse it in the model. In addition, it is worth noticing that DUCK exceeds the results of generative models like *GENRE* (De Cao et al., 2021) and *CM3-Medium*. This is impressive considering that generative models are notoriously more expensive than bi-encoder models and require one order of magnitude more time per mention at inference (Ayoola et al., 2022). Finally, we see that DUCK meets the performance of *CM3-Large* (Aghajanyan et al., 2022), a generative model that, with its 13 billion parameters, is almost 5 times larger than *CM3-Medium* (2.7 billion parameters) and more than 18 times larger than DUCK (717 million parameters).

**Knowledge-aware methods.** DUCK uses a knowledge graph (Wikidata) to infuse additional information in the model. While some methods listed in Table 1 use indeed type information extracted from Wikidata (Ayoola et al., 2022; Orr et al., 2021; Chen et al., 2020), other existing knowledge-aware methods for entity disambiguation have reported results in different experimental settings, evaluating on **AIDA**, with the candidate set of Pershina et al. (2015). In order to compare with these methods, we evaluated DUCK on the candidate set of Pershina et al. (2015), and we report the results in Table 2. Interestingly, our model outperforms both *DeepType* (Raiman and Raiman, 2018) and the methods of (Mulang et al., 2020) and Onoe and Durrett (2020). State-of-the-art results in this

| Method | AIDA |
|---|---|
| Onoe and Durrett (2020) | 85.9 |
| Raiman and Raiman (2018) | 94.9 |
| Mulang et al. (2020) | 94.9 |
| Ayoola et al. (2022) (*Wikipedia*) | 89.1 |
| Ayoola et al. (2022) (*fine-tuned*) | **97.1** |
| DUCK (*Wikipedia*) | 94.3 |
| DUCK (*fine-tuned*) | 96.4 |

Table 2: Micro-$F_1$ (*InKB*) results of knowledge-aware methods on the candidate set of Pershina et al. (2015).

| Method | AIDA | MSNBC | AQUAINT | ACE2004 | CWEB | WIKI | Avg. |
|---|---|---|---|---|---|---|---|
| DUCK w/o types (*Wikipedia*) | 85.0 | 93.1 | 87.5 | 87.5 | 73.6 | 84.5 | 85.2 |
| DUCK cartesian coord. (*Wikipedia*) | 90.6 | 94.9 | 91.3 | 95.0 | 76.5 | 85.1 | 88.9 |
| DUCK w/o candidate set (*Wikipedia*) | 87.4 | 89.9 | 85.2 | 88.8 | 69.1 | 82.0 | 83.7 |
| DUCK (*Wikipedia*) | **91.0** | **95.1** | **91.3** | **95.4** | **76.9** | **86.1** | **89.3** |

Table 3: Micro-$F_1$ results achieved by several ablations of DUCK

setting are obtained by Ayoola et al. (2022), confirming the overall good performance obtained by this method on **AIDA** in Table 1. However, notice that *(a)* Ayoola et al. (2022) trained the model on a custom Wikipedia dump, consisting of 100M mention-entity pairs (more than one order of magnitude larger than our dataset) and *(b)* DUCK obtains excellent results even in an out-of-domain scenario (without fine-tuning on AIDA), reaching 94.3 micro-$F_1$ points (an improvement of 5.1 points with respect to Ayoola et al., 2022).

## 5.3 Ablation studies

In order to provide more insights into the performance of the model, we performed several ablation studies. First, we performed an ablation where we removed the contribution of the $\mathcal{L}_{Duck}$ terms and the L2 regularization $l_2$ from the loss function (DUCK w/o types). In this case, we only train the model using the entity-disambiguation loss $\mathcal{L}_{ED}$, without infusing any type information. In addition, we assessed the benefit of using box embeddings in spherical polar coordinates by experimenting with a version of the model where boxes are expressed in cartesian coordinates (DUCK cartesian coord). In this case, we parametrize a box as a pair of vectors $Box(r) = (\boldsymbol{r}^-, \boldsymbol{r}^+)$, where

$$\boldsymbol{r}^- = \text{FFN}^-(\boldsymbol{r}),$$
$$\boldsymbol{r}^+ = \boldsymbol{r}^- + \text{ReLU}(\text{FFN}^+(\boldsymbol{r})) + \delta'_{\min}.$$

As before, $\delta'_{\min}$ is a margin parameter that defines the minimum width of a box, $\text{FFN}^-$ and $\text{FFN}^+$ are feed-forward networks, and $\text{ReLU}(x) = \max(0, x)$ is the ReLU activation function. Finally, we report the results obtained by DUCK when no candidate set is provided (DUCK w/o candidate set). In this case, we score each mention against the whole set of entities (which amounts to almost 6M entities). Table 3 shows the results achieved by the ablations described above. Including entity types boosts the performance by approximately 4 micro-$F_1$ points and up to 7.9 points on **ACE2004**. The results further show the benefit of using spherical coordinates and that the model achieves good performance even without a candidate set.

## 5.4 Qualitative analyses

This section complements the quantitative results discussed so far with some qualitative analyses.

**Analysis of the boxes.** Table 4 shows a qualitative analysis of the relative placement of entities and boxes in the latent space. In the left side of the table, we looked into three boxes corresponding to the relations *flag*, *sport*, and *director*, and we reported the top 10 entities that are closer to the center of the box. The examples show a clear clustering of types, as all entities closer to the box *flag* are countries, entities inside the box *sport* are sport teams, and entities inside the box *director* are movies. For the latter box, we observe that the model predicts two movies that, in Wikidata, are missing the relation *director*, showing the ability of the model to robustly deal with incomplete information. In the right side of Table 4, we show which boxes are closer to three entities, namely a country, a football team and a movie, according to the distance function defined in Section 3. The examples show that the model is able to correctly place entities of different types in different boxes.

**Examples.** Figure 3 shows examples of the entities predicted by our method, for inputs where the prediction of DUCK differs from the ablation that does not use type information. The first two examples (left and center), clearly show how type information can help the disambiguation in cases where some keywords in the context of the mention misleads the model to making a wrong prediction. The third example (right) shows a case where DUCK predicts correctly the type of the mention, but fails to leverage some contextual information and links it to a wrong entity. This is likely due to the two entities sharing most boxes and being very close in the embedding space.

| Flag | Sport | Director | Italy | Italy national football team | Fantastic Beasts and Where to Find Them |
|---|---|---|---|---|---|
| Czech Republic | The Invincibles (en. football) | A Secret Life (film) | country | sport | orig. lang. of film or TV show |
| France | New England Tea Men | The Dream (1989 film) | GeoNames ID | head coach | genre |
| Austria | EMKA Racing | The Prodigal (1983 film) | flag image | country | publication date |
| Poland | Los Angeles Heroes | A Time to Sing (film) | legislative body | inception | form of creative work |
| Sweden | Hamline Pipers football | Modern Romance (film) | cat. for ppl. born here | cat. for memb. of team | language of work or name |
| Mexico | La Máquina | A Sinful Life | locator map image | Facebook ID | main subject |
| Germany | A. J. Foyt Enterprises | The Devil's Arithmetic (film) | Commons gallery | owned by | distributed by |
| D. R. of Congo | Atlético Minero | Unchained (film) | shares border with | country for sport | director |
| Italy | Artiach (cycling team) | As Is (film) | continent | league | cast member |
| Argentina | PS Barito Putera U-20 | Identity Crisis (film) | named after | social media followers | spoken text audio |

Table 4: Qualitative analysis of the boxes predicted by DUCK. The left side of the table shows the closest entities to a box, ranked according to the distance function defined in Section 3. The right part of the table shows the closest boxes to a given entity. Correct predictions are highlighted in green, whereas predictions that do not match relations in Wikidata are highlighted in red. Best viewed in color.

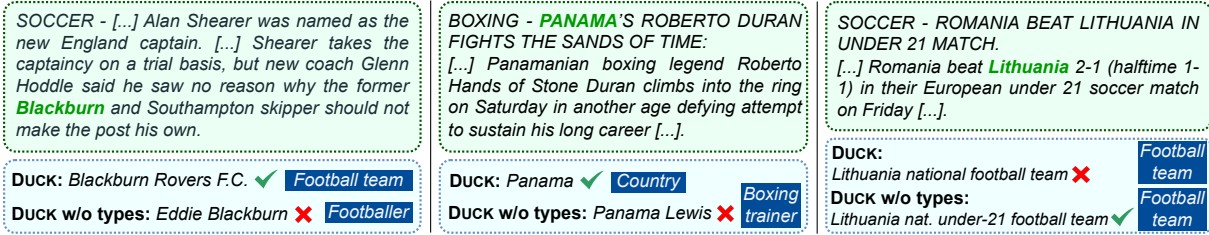

Figure 3: Examples of the predictions of **DUCK** and **DUCK w/o types**, showing cases where **DUCK** predicts the correct entity (left and center) and where it predicts a wrong one (right). Mentions are highlighted in **bold green**.

## 6 Related work

Our work builds on top of the bi-encoder architecture of Wu et al. (2020) and was partially motivated by the work of Raiman and Raiman (2018), who showed the benefit of using type information for entity disambiguation. Previous research has employed a variety of methods to model mentions and entities using neural networks (He et al., 2013; Sun et al., 2015; Yamada et al., 2016; Kolitsas et al., 2018). Our method falls within a recent line of work that has proposed approaches to use type information in the disambiguation process (Raiman and Raiman, 2018; Khalife and Vazirgiannis, 2019; Onoe and Durrett, 2020; Chen et al., 2020; Orr et al., 2021; Ayoola et al., 2022). The closest method to DUCK is the one of Ayoola et al. (2022), who incorporated type knowledge in a bi-encoder model similar to the one of Wu et al. (2020). The main difference between DUCK and the model of Ayoola et al. (2022) is that they used type labels extracted from Wikidata instead of our duck-typing approach, they represented types as points in the latent space, and they improved the performance of the model by using global entity priors (i.e., prior probabilities of an entity given a mention) extracted from count statistics. Broadly speaking, our method falls within the scope of recent research to infuse prior knowledge in neural models (Lake et al., 2017; Atzeni et al., 2023). Several methods have been proposed to achieve this goal, like infusing commonsense knowledge extracted from KGs in attention-based models (Bosselut et al., 2019; Murugesan et al., 2021b,a), constraining attention weights in transformers using graph-structured data (Sartran et al., 2022), and improving reasoning abilities of language models with graph neural networks (Yasunaga et al., 2021; Atzeni et al., 2021).

## 7 Conclusion

This paper introduced DUCK, a method to improve the performance of entity disambiguation models using prior type knowledge. The overall idea underlying our method was inspired by the concept of duck typing, as we defined types in a fuzzy manner, without any need for type labels. We introduced box embeddings in spherical polar coordinates and we demonstrated that using this form of representation allows effectively clustering entities of the same type. Crucially, we showed that infusing structural information in the latent space is sufficient to close the gap between efficient methods based on dense retrieval and generative models. As a future line of research, it might be interesting to explore methods to infuse prior knowledge of entity types in generative models as well.

## Limitations

Our method assumes that we have access to both entity descriptions in natural language and a knowledge graph providing relations between pairs of entities. Methods based on dense retrieval (without type information) usually rely only on the first assumption. In our experiments, entity descriptions are obtained from Wikipedia (more precisely, from the KILT dump of Petroni et al., 2021) and we rely on Wikidata (Vrandečić and Krötzsch, 2014) as the underlying KG. In domain-specific applications, one of the two sources of information (typically the KG) might not be available. However, notice that all existing type-aware methods have a similar limitation, as they require type labels at training time. We believe that other forms of structured knowledge might be used in some cases to obtain the attributes needed to represent the type of an entity. Also, we point out that training on Wikipedia is a very common choice and many real-world applications rely on the same setup employed in this paper (Plekhanov et al., 2023; Ayoola et al., 2022).

Compared to other type-aware methods, DUCK has the disadvantage that we cannot predict the type of a mention in the form of a label. This is a design choice that allows modeling type information in a more fine-grained manner. As shown in the paper, this choice results in better overall entity-disambiguation performance compared to other type-aware methods. In applications where it would be interesting to obtain the type of a mention in the form of a label, we believe that a simple heuristic correlating the relations of an entity to its type in Wikidata would be very effective. We refer the reader to Appendix A for insights on the type information carried by relations in a KG.

Additionally, we emphasize that the choice of spherical polar coordinates for modeling relational information is dependent on the use of the dot product or the cosine similarity as the function for ranking the closest entities to a given mention. In case a different function is used (e.g., the $L_2$ distance), then box embeddings in cartesian coordinates might be better suited. We used the dot product because it is the most popular choice, allowing us to build on the model of Wu et al. (2020).

One more caveat is that our method is sensitive to the margin parameter $\gamma$. In case DUCK is trained on different domains, it might be beneficial to tune this parameter carefully. We tried using probabilistic box embeddings (similar to Dasgupta et al., 2020 and Li et al., 2019) in order to get rid of the margin parameter and optimize the model using a cross-entropy loss, but we obtained better results with the method described in the paper.

Finally, our loss function does not optimize only for entity disambiguation. Hence, DUCK might occasionally loose contextual information, in favor of placing the mention in the correct boxes. This is shown in the rightmost examples of Figure 3 and Figure 5. In both cases, the correct entity and the prediction have lexically similar descriptions and share many relations. Therefore, their embeddings are close, making the disambiguation task more difficult. We noticed empirically that, when this happens, the model might be biased towards the more common entity. For instance, in the example of Figure 3, DUCK predicts the main national team over the under-21 team, whereas in the example of Figure 5, the model favors the football team over the basketball one.

## Ethical considerations

Entity disambiguation is a well-known task in natural language processing, with several real-world applications in different domains, including content understanding, recommendation systems, and many others. As such, it is of utmost importance to consider ethical implications and evaluate the potential bias that ED models could exhibit. DUCK is trained on Wikipedia and Wikidata (Vrandečić and Krötzsch, 2014), which can carry bias (Sun and Peng, 2021). These biases may be related to geographical location (Kaffee et al., 2018; Beytía, 2020), gender (Hinnosaar, 2019; Schmahl et al., 2020), and marginalized groups (Worku et al., 2020). Since at inference time DUCK is essentially a bi-encoder architecture based on dense retrieval, the index of entity embeddings can be updated without retraining the model. This would allow incorporating efforts to reduce bias in Wikipedia efficiently. Another source of bias in DUCK (and in many downstream tasks in natural language processing) is the underlying pre-trained language model used to initialize the entity and mention encoders. In our experiments, we used the RoBERTa model of Liu et al. (2019). Steed et al. (2022) show that bias mitigation needs to be performed on the downstream task directly, rather than on the language model. We refer the reader to Rudinger et al. (2018) and Zhao et al. (2018) for methods to mitigate bias in downstream tasks.

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

## A  Duck typing on knowledge graphs

To get more insights into our definition of duck typing on knowledge graphs, we performed a qualitative analysis of entities that share a large number of relations in Wikidata (Vrandečić and Krötzsch, 2014). Precisely, we used the cardinality of the symmetric difference between the sets of relations of two entities $e_1, e_2 \in \mathcal{E}$, defined as:

$$
\begin{aligned}
dist_{KG}(e_1, e_2) &= |\mathcal{R}(e_1) \triangle \mathcal{R}(e_2)| \\
&= |(\mathcal{R}(e_1) \setminus \mathcal{R}(e_2)) \cup (\mathcal{R}(e_2) \setminus \mathcal{R}(e_1))| \\
&= |(\mathcal{R}(e_1) \cup \mathcal{R}(e_2)) \setminus (\mathcal{R}(e_1) \cap \mathcal{R}(e_2))|
\end{aligned}
$$

as a measure of the distance between the types of two entities $e_1$ and $e_2$. Notice that the distance defined above can be expressed as the Hamming distance between binary encodings of the sets of relations, hence we can efficiently retrieve the neighbors of a given entity on GPU, following the method of Johnson et al. (2021). If our definition of duck typing works well, we expect entities with low distance to be likely of the same type. Table 5 shows the top-10 neighbors that minimize the distance function defined above for several entities. We emphasize that these lists of neighbors are not produced by our model, rather they are examples of the prior knowledge that we aimed to infuse in DUCK. This analysis shows that our notion of duck typing carries fine-grained type information, as it allows detecting countries, cities, highly influential computer scientists and mathematicians, football players, singers, politicians, animals, companies, scientific awards and more.

## B  Entity-box distance

This section provides more details on the entity-box distance function defined in Section 3.2. A plot of the distance function in the uni-dimensional case, for a scalar entity representation $e$ and several boxes centered at $\pi/2$ with different scalar widths $\delta_r$ is shown in Figure 4. The plot shows that the distance function has different slopes for entities inside and outside the boxes. This is meant to strongly penalize entities that lie outside boxes corresponding to their relations, as it ensures that outside points receive high gradient through which they can more easily reach their target box. Additionally, recalling the expression for $dist(e, r)$ given in Section 3.2, notice that the distance depends on the width of the box. More precisely, whenever an entity is inside its target box, the distance inversely

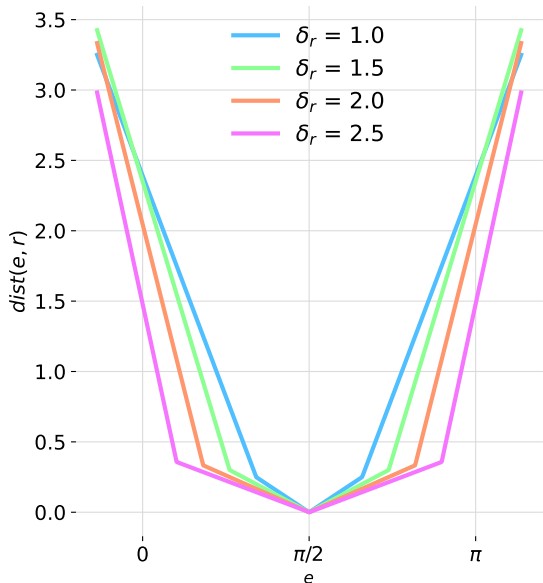

Figure 4: Plot of the entity-box distance in the uni-dimensional case, for boxes centered at $\pi/2$ with different scalar widths $\delta_r$

correlates with box size. This allows maintaining low distance values inside large boxes while providing a gradient to keep points inside. For entities outside their target boxes, the distance linearly correlates with the width of the box, to penalize points outside larger boxes more severely.

## C  Details on the model

In order to train DUCK, we need to convert the entity representations $\boldsymbol{e}$ into spherical polar coordinates (the same applies to the mention representations $\boldsymbol{m}$). This can be done as follows:

$$
\varphi_{e,1} = \arccos\left(\frac{e_1}{\sqrt{e_d^2 + e_{d-1}^2 + \cdots + e_1^2}}\right)
$$

$$
\varphi_{e,2} = \arccos\left(\frac{e_2}{\sqrt{e_d^2 + e_{d-1}^2 + \cdots + e_2^2}}\right)
$$

$$
\vdots
$$

$$
\varphi_{e,d-2} = \arccos\left(\frac{e_{d-2}}{\sqrt{e_d^2 + e_{d-1}^2 + e_{d-2}^2}}\right)
$$

$$
\varphi_{e,d-1} = \begin{cases} \arccos\left(\frac{e_{d-1}}{\sqrt{e_d^2 + e_{d-1}^2}}\right) & \text{if } e_d \geq 0 \\ 2\pi - \arccos\left(\frac{e_{d-1}}{\sqrt{e_d^2 + e_{d-1}^2}}\right) & \text{if } e_d < 0 \end{cases}
$$

| Italy | London | Rome | Alan Turing | Ada Lovelace |
|---|---|---|---|---|
| Portugal | Istanbul | Milan | Bernhard Riemann | Lady Byron |
| Spain | Madrid | Florence | Kurt Gödel | Rosalind Franklin |
| Norway | Istanbul Province | Naples | John von Neumann | Elizabeth Fry |
| Greece | Stockholm | Venice | Herbert A. Simon | Catherine Dickens |
| Poland | Cairo | Turin | John Forbes Nash Jr. | Rosina Bulwer Lytton |
| Denmark | Buenos Aires | Palermo | Claude Shannon | Lady Emmeline Stuart-Wortley |
| Belgium | Manchester | Rio de Janeiro | Nikolai Lobachevsky | Eleanor Marx |
| Hungary | Amsterdam | Genoa | Benoit Mandelbrot | Wilhelmina Powlett, Duchess of Cleveland |
| Finland | Milan | Bologna | Willard Van Orman Quine | Rachel Russell, Lady Russell |
| Republic of Ireland | Ankara | Lisbon | Niels Henrik Abel | Martha Jefferson |
| **Cristiano Ronaldo** | **Justin Bieber** | **Donald Trump** | **Lion** | **Jaguar** |
| Lionel Messi | Harry Styles | Joe Biden | Tiger | Cougar |
| Luis Suárez | Chris Brown | Mrs. Bill Clinton | Leopard | Ocelot |
| Gerard Piqué | Ed Sheeran | Barack Obama's | Cheetah | Giant anteater |
| Neymar | Eminem | George W. Bush | Jaguar | Giant armadillo |
| Manuel Neuer | Camila Cabello | Al Gore | Red panda | Chimpanzee |
| Paul Pogba | Nick Jonas | Kamala Harris | Giant panda | Bonobo |
| Ronaldinho | Jordin Sparks | Elizabeth Warren | Cougar | Indian rhinoceros |
| Luka Modrić | Richard Marx | Bill Clinton | Okapi | Giant otter |
| Antoine Griezmann | Niall Horan | Michael Bloomberg | Hippopotamus | Black rhinoceros |
| Gareth Bale | Shawn Mendes | Benjamin Netanyahu | Fennec fox | Pronghorn |
| **Jaguar Cars** | **Maserati** | **Veliko Tarnovo** | **Gracilinanus** | **Fields Medal** |
| Land Rover | Lancia | Kluczbork | Scolomys | IEEE Medal of Honor |
| Steyr-Daimler-Puch | VinFast | Yambol | Aethalops | Kavli Prize |
| MG Cars | McLaren Automotive | Kyustendil | Oligoryzomys | Rosenstiel Award |
| Gulf Oil | Massimo Dutti | Targovishte | Raphicerus | Paul Ehrlich and Ludwig Darmstaedter Prize |
| British Motor Corporation | Paper Mate | Pančevo | Vesper mouse | Albert Einstein World Award of Science |
| Safeway (UK) | Infiniti | Kragujevac | Rhabdomys | Canada Gairdner International Award |
| Rover Group | Peroni Brewery | Ruse, Bulgaria | Balantiopteryx | Earle K. Plyler Prize for Molecular Spectroscopy |
| Rover Company | Lotus Cars | Sombor | Arielulus | Dannie Heineman Prize for Mathematical Physics |
| MIPS Technologies | Colruyt (supermarket) | Vratsa | Bassariscus | Bôcher Memorial Prize |
| F. W. Woolworth Company | Overkill Software | Pazardzhik | Microsciurus | NAS Award in Chemical Sciences |

Table 5: Top 10 entities with the most similar set of relations to a given entity in Wikidata

where $e_i$ is the $i$-th entry of the entity representation $\boldsymbol{e} \in \mathbb{R}^d$ and $\varphi_{e,i}$ is the $i$-th component of the representation in spherical coordinates $\boldsymbol{\varphi}_e \in \mathbb{R}^{d-1}$. Looking at the equations above, we notice that in a spherical coordinate systems, all angles range from 0 to $\pi$, with the only exception of the last coordinate $\varphi_{e,d-1}$, which ranges from 0 to $\pi$ if $e_d$ is positive and from $\pi$ to $2\pi$ otherwise. In order to make the definition of the boxes and of the entity-box distance simpler, we decided to constrain the last coordinate in the range $[0, \pi]$ as well. We achieved this objective by constraining the last coordinate of the entity and mention representations to be positive, applying an absolute value to the last dimension of the output of the entity and mention encoders. This essentially restricts all representations and boxes to be on half of the hypersphere, more precisely on the portion where $e_d > 0$. We apply this transformation before computing the overall optimization objective of Section 3.2.

## D   Training details

In order to train DUCK, we need to select negative entities $e_j$ for the entity-disambiguation loss $\mathcal{L}_{ED}$ of Section 3. Having high-quality negative entities is crucial to achieve high performance, thus we trained DUCK in several stages.

First, we trained the model using, as negative entities for each mention, all entities in the same batch. In order to provide more meaningful information, we further added entities that maximize a prior probability $\hat{p}(e|m)$, extracted from count statistics derived from large text corpora. In details, we used the prior probabilities of Ayoola et al. (2022). Notice that, differently from Ayoola et al. (2022), we do not use these prior probabilities at inference time, but we only use them to provide better negative entities to the model in this first training stage. For each mention, we included in a batch 3 negative entities that maximize the prior probability, limiting the total number of entities in a batch to 32 (per GPU). In this stage, we use a batch size of 16 mentions (per GPU), which means that, for some mentions, we do not have the entities that maximize the prior probability. This is not an issue, as we still use all entities in the same batch as negatives for every mention. To compute the loss $\mathcal{L}_{Duck}$, we used a sampling temperature of $\alpha = 0.1$. Furthermore, in order to provide a better training signal and counteract missing information in the knowledge graph, we only trained the model using entities that have at least 5 relations.

We trained the model for 1 epoch on 8 GPUs, validating on the BLINK validation set every 5000

| Member of political party | Member of sports team | Cast member |
|---|---|---|
| Richard Nixon | Justin Moore (soccer) | A Time to Sing (film) |
| Édouard Philippe | Scott Jones (Puerto Rican footballer) | A Secret Life (film) |
| Kaname Tajima | Blake Camp | The Dream (1989 film) |
| Laurent Fabius | Miles Robinson (soccer) | Can You Hear the Laughter? The Story of Freddie Prinze |
| Albert II, Prince of Monaco | Simon Thomas (soccer) | I Was a Teenage TV Terrorist |
| Joe Biden | Scott Wilson (footballer, born 1993) | A Sinful Life |
| François Hollande | Scott Jenkins (soccer) | The Morning After (1986 film) |
| Jean-Marc Ayrault | Ali Mohamed (footballer) | Cries Unheard: The Donna Yaklich Story |
| Ursula von der Leyen | Scott Fraser (footballer, born 1995) | My Sex Life... or How I Got into an Argument |
| Yasutomo Suzuki | Justin Willis (soccer) | Enemies, A Love Story (film) |

Table 6: Closest Wikipedia entities to different boxes according to the entity-box distance function. Correct predictions are highlighted in green, whereas predictions that do not match relations in Wikidata are highlighted in red. Best viewed in color.

| Member of political party | Member of sports team | Cast member |
|---|---|---|
| Ronald Reagan | Tayfun Korkut | The Crying Game |
| Jacques Chirac | Lilian Thuram | Ice Cold in Alex |
| George H. W. Bush | Scott Sanders (baseball) | Michael Collins (film) |
| Madeleine Albright | Roberto Carlos (footballer) | Viva Zapata! |
| Deng Xiaoping | Tony Adams (footballer) | On the Waterfront |
| Bob Dole | Stuart McCall | Lawrence of Arabia (film) |
| Masoud Barzani | Joakim Persson | My Turn (memoir) |
| Yasser Arafat | Cosmin Contra | Aidan Quinn |
| Bill Clinton | Todd Martin | Der Spiegel |
| Liam Neeson | Geoff Aunger | Julia Roberts |

Table 7: Closest entities (extracted from the validation set of the **AIDA** dataset) to different boxes according to the entity-box distance function. Correct predictions are highlighted in green, whereas predictions that do not match relations in Wikidata are highlighted in red. Only 6 entities in **AIDA** have the relation *Cast member* and the model is able to correctly retrieve all of them, has shown above. Best viewed in color.

gradient steps. Then, we used the model that maximizes the validation performance to produce a representation for every entity, and we mined the closest representations for each entity in Wikipedia. This step is usually referred to as hard-negative mining. We used these entities as negative examples for the $\mathcal{L}_{ED}$ loss and trained the model again, starting from the same checkpoint employed for the negative-mining stage. We used a batch size of 16, with 3 negative examples for each mention and up to 32 entities in a batch. We increased the sampling temperature for the boxes to $\alpha = 0.5$, keeping a threshold of at least 5 relations for each entity. We trained the model for one more epoch, validating every 5000 gradient steps as before.

Finally, we repeated the hard-negative mining process and kept training the model for 10 000 additional gradient steps, using a batch size of 4, 5 hard negatives for each mention and up to 3 entities that maximize the prior probability $\hat{p}(e|m)$ (if distinct from the negatives). As we increased the number of negative entities, the batch size is significantly smaller than before. Therefore, we did

gradient accumulation for 4 steps. Furthermore, we increased the maximum length of a mention from 128 tokens to 512, and set the sampling temperature to $\alpha = 1.0$. In this final stage, we assumed the model had already learned to place entities in their target boxes, hence we used all entities in the dataset, regardless of the number of relations they have in Wikidata.

# E   Additional results

Table 8 provides additional results obtained by the ablations of DUCK described in Section 5.3, after fine-tuning on **AIDA**. For reference, we report the results of the main model as well. Overall, the experiments confirm what we observed in Section 5.3. The main model performs consistently better than all ablations across all datasets except **ACE2004**, where the model in cartesian coordinates obtains the same results achieved by DUCK (*Wikipedia*) in Table 1. Overall, we notice that infusing type information using *duck typing* improves downstream performance and using polar coordinates is beneficial over boxes in cartesian coordi-

| Method | AIDA | MSNBC | AQUAINT | ACE2004 | CWEB | WIKI | Avg. |
|---|---|---|---|---|---|---|---|
| DUCK w/o types (*fine-tuned*) | 89.1 | 92.5 | 87.4 | 87.1 | 74.9 | 83.8 | 85.8 |
| DUCK cartesian coord. (*fine-tuned*) | 92.1 | 94.0 | 90.6 | **95.4** | 77.5 | 85.5 | 89.2 |
| DUCK w/o candidate set (*fine-tuned*) | 90.9 | 90.5 | 86.3 | 89.2 | 71.1 | 81.9 | 85.0 |
| DUCK (*fine-tuned*) | **93.7** | **94.6** | **91.3** | 95.0 | **78.2** | **85.9** | **89.8** |

Table 8: Micro-$F_1$ results achieved by several ablations of DUCK after fine-tuning on **AIDA**

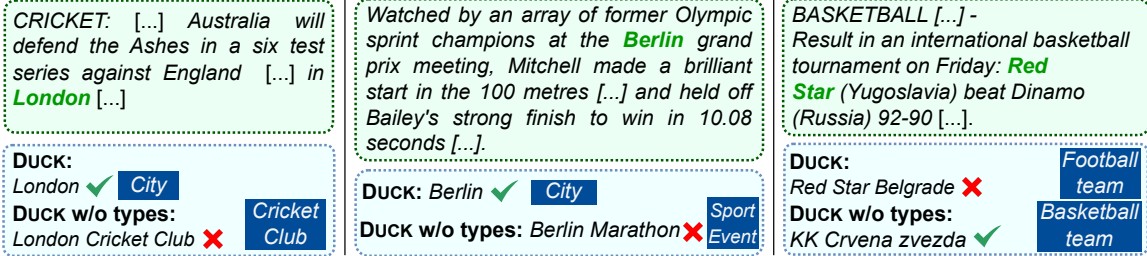

Figure 5: Further examples of the predictions of **DUCK** and **DUCK w/o types**. Mentions highlighted in **bold green**.

nates. Interestingly, the model is able to achieve a $F_1$ score of 85.0 even without the candidate set, confirming the intuition that using type information to structure the latent space is advantageous for the entity-disambiguation task.

## F  Additional qualitative results

Following the qualitative analyses of Section 5.4, in this section we provide additional results and further examples.

**Analysis of the boxes.** Table 6 shows additional examples of the top-10 entities lying closer to the center of a box. This analysis is performed on all entities in Wikipedia (approximately 6M entities for the English language) and complements the examples reported on the left side of Table 4. In this case, we analyzed three more relations, namely *member of political party*, *member of sports team*, and *cast member*. The model correctly reports politicians for the first box, athletes for the second, and movies for the latter, confirming the clustering of entity types that we noticed in Section 5.4. Additionally, the model appears robust to missing information in the knowledge graph, being able to predict the relation *cast member* for movies that are missing it in the KG.

We performed the same analysis, using the same set of relation, on the entities appearing in the validation set of the **AIDA** dataset. The results are reported in Table 7. Since **AIDA** contains news articles, the dataset includes several mentions of politicians and athletes, and the model is able to correctly cluster the two types of entities (with only

one error in the top 10 predictions for the relation *member of political party*). On the other hand, the dataset includes only 6 entities that are movies (more precisely, entities with the relation *cast member*). Interestingly, the top-10 entities closer to the center of the box corresponding to the relation *cast member* are all the movies mentioned in AIDA. The remaining 4 entities listed in Table 7 include two actors (*Aidan Quinn* and *Julia Roberts*), suggesting that the embedding space carries semantic information and that actors are closer to movies than other entities.

**Examples.** Figure 5 shows further examples of the predictions of DUCK and the ablation **DUCK w/o types**. Confirming the insights of Figure 3, the first two examples (left and center), show that DUCK is usually able to predict entities of the correct type and how this can help the model in making the correct prediction. The third example (right) shows a case where the model predicts a wrong entity, as it links the mention to a football team, though the context clearly suggests that the correct entity should be a basketball team instead. This suggests that, in some rare cases, DUCK might give too much weight to the prior knowledge about the relations of candidate entities, loosing knowledge coming from the description of the entity and from contextual information about the mention.

## G  Hyperparameters and reproducibility

We trained DUCK using the AdamW optimizer (Loshchilov and Hutter, 2019) on 8 NVIDIA A100 GPUs, each with 40 GB of memory. Following

Ayoola et al. (2022), we initialized the learning rate to 0 and linearly increased it up to $1.00 \times 10^{-5}$ over the first 5000 steps. To avoid catastrophic forgetting, we set a maximum learning rate of $1.00 \times 10^{-6}$ when fine-tuning on **AIDA**. We limited the number of entities in a batch (which are used to compute the loss term $\mathcal{L}_{ED}$) to 32 per GPU, but we shared entity representations across all devices when computing the loss, reaching an effective maximum number of entities of $32 \times 8 = 256$ per batch. We increased the maximum length of a mention to 512 tokens at inference time. Table 9 reports the values of all hyperparameters of the model for reproducibility of our results.

| Hyperparameter | Value |
|---|---|
| Learning rate (max) | $1.00 \times 10^{-5}$ |
| Learning rate warm-up steps | 5000 |
| $\gamma$ | 2 |
| $\lambda_{Duck}$ | 0.1 |
| $\lambda_{l_2}$ | 0.1 |
| Number of negative boxes $k$ | 512 |
| $\delta_{\min}$ | 0.1 |
| $\delta'_{\min}$ | 0.1 |
| $d$ | 1024 |
| $\alpha$ | See Appendix D |
| Max entity length $n_e$ | 128 |
| Max mention length $n_m$ | See Appendix D |
| Max relation length $n_r$ | 256 |
| Batch size | See Appendix D |
| Max num. entities per batch (per GPU) | 32 |

Table 9: Hyperparameter values of DUCK