# OpenReview forum: "Polar Ducks and Where to Find Them: Enhancing Entity Linking with Duck Typing and Polar Box Embeddings"
_EMNLP/2023/Conference — EMNLP 2023 Main_

### Official Review · Reviewer_F4hB · 2023-08-04

**Soundness:** 4

**Excitement:**

4: Strong: This paper deepens the understanding of some phenomenon or lowers the barriers to an existing research direction.

**Paper Topic And Main Contributions:**

The paper is devoted to improving entity linking using polar box embeddings and relation encoding. The basic approach is based on dense retrieval. To improve entity matching, special encoding of relation sets is aded. The approach is tested on several datasets and shows average improvement in performance.

**Reasons To Accept:**

1. New type of embeddings --  box embeddings in polar coordinates is proposed.
2. The proposed embeddings are  used for representing similarities of entities according to sets of their relations.
3. The approach allows improving the results of entity linking over strong state-of-the art results.


**Reasons To Reject:**

1. The approach is quite complex and difficult for reproduction. In the paper there are no promices to publish the code,
2.  The method requires much information about an entity:  entity description from Wikipedia and the entity relation set from Wikidata. Such information can be not accessible in real tasks.

**Reproducibility:**

2: Would be hard pressed to reproduce the results. The contribution depends on data that are simply not available outside the author's institution or consortium; not enough details are provided.

**Reviewer Confidence:**

4: Quite sure. I tried to check the important points carefully. It's unlikely, though conceivable, that I missed something that should affect my ratings.

---

> ### Author Rebuttal · Authors · 2023-08-28
>
> We thank the reviewer for the positive feedback and the accurate review. We provide answers to your main comments below.
>
> **Reproducibility**. **We will publish the source code upon acceptance of the paper**, together with a clear documentation explaining how to reproduce our results. We strived to include in the paper all details on the infrastructure used for our experiments and all hyperparameters in Appendix G. We further reported details on how the model was trained in Appendix D. All datasets used to train the model and evaluate it are publicly available and consistent with prior work. We downloaded the datasets from the [repository](https://github.com/facebookresearch/GENRE/blob/main/scripts_genre/download_all_datasets.sh) of De Cao et al. (2021).
>
> **Requirement for relational information.** We agree that the requirement for both textual descriptions and a knowledge graph is a limitation of our method, as mentioned in the limitations section on page 9. Notice that all existing type-aware methods we are aware of, including Ayoola et al. (2022), Orr et al. (2021) and Chen et al. (2020), have a similar limitation (even stricter, as they require type labels). However, we point out that, in real-world applications, training on Wikipedia is a very popular choice due to the large coverage of this dataset. Several industrial applications rely on the same training setup described in this paper and can take advantage of the availability of both textual descriptions and a knowledge graph (Plekhanov et al. 2023, Botha et al., 2020; Ayoola et al., 2022).

---

### Official Review · Reviewer_89ts · 2023-08-04

**Soundness:** 4

**Excitement:**

4: Strong: This paper deepens the understanding of some phenomenon or lowers the barriers to an existing research direction.

**Paper Topic And Main Contributions:**

In this proposal, a novel approach for entity linking is presented, called Ducks. This method utilizes a combination of relations from a knowledge graph and context to determine entity types. The entity's relations are represented using box embeddings in spherical polar coordinates. The mention and entity are optimized to position entities with similar types within the corresponding boxes of their relations.

**Questions For The Authors:**

- Given an entity, could you provide more details about relation modelling and relation encoder?

**Reasons To Accept:**

- The paper addresses a critical problem and develops an intuitive and sensible in-depth analysis.
- The paper is very well written, including nice clear examples, and appropriately discusses the state of the art.

**Reasons To Reject:**

- I have an overall positive impression, except for the fact that the paper may still miss some discussions about how relations are  modelled and extracted from Wikipedia.

**Reproducibility:**

3: Could reproduce the results with some difficulty. The settings of parameters are underspecified or subjectively determined; the training/evaluation data are not widely available.

**Reviewer Confidence:**

3: Pretty sure, but there's a chance I missed something. Although I have a good feel for this area in general, I did not carefully check the paper's details, e.g., the math, experimental design, or novelty.

---

> ### Author Rebuttal · Authors · 2023-08-28
>
> We thank the reviewer for the positive comments and the useful feedback. We address your main questions below.
>
> **Relation modeling.** We model relational information as follows. For each entity in Wikipedia, we look for the node corresponding to that entity in the Wikidata knowledge graph. Then, for each edge originating from that entity, we consider the relation labeling that edge and we model it using its label and description in English.
>
> As an example, the entity “_Italy_” in Wikipedia corresponds to node _Q38_ in Wikidata. We used the mapping from Wikipedia titles to Wikidata identifiers in De Cao et al. (2021) (https://github.com/facebookresearch/GENRE). Node _Q38_ has several relations (see https://www.wikidata.org/wiki/Q38) each with a label and a description. For instance, one of these relations is property _P1082_ (https://www.wikidata.org/wiki/Property:P1082), whose English label is “_population_” and its description is “_number of people inhabiting the place; number of people of subject_”. We encode the label and description of a relation using a pre-trained RoBERTa model and we keep these embeddings frozen at training time. The relation encoder then is a feed-forward network that produces the corners of the boxes based on the pre-trained relation embeddings with the approach mentioned in Section 4.2. We will clarify these details in the final version of the paper.
>
> **Reproducibility.** As a note on reproducibility, we strived to include in the paper all details on the infrastructure used for our experiments and all hyperparameters in Appendix G. We further reported details on how the model was trained in Appendix D. All datasets used to train the model and evaluate it are publicly available and consistent with prior work. We downloaded the datasets from the [repository](https://github.com/facebookresearch/GENRE/blob/main/scripts_genre/download_all_datasets.sh) of De Cao et al. (2021). **We will open source the code of our method upon acceptance of the paper**.

---

### Official Review · Reviewer_VEWh · 2023-08-05

**Soundness:** 4

**Excitement:**

4: Strong: This paper deepens the understanding of some phenomenon or lowers the barriers to an existing research direction.

**Paper Topic And Main Contributions:**

The authors solve the Entity Disambiguation task by applying Duck typing and polar box embeddings. The authors represent entity types via relations, so that entities that share many relations are clustered together as the same type. Then the authors apply a bi-encoder model to a pair of a mention and a candidates entity and combine the loss function with the loss function of duck typing.

**Reasons To Accept:**

The paper is well-written and easy to follow. The applied methodology and the formulation of the task is clear and the limitations and the reason of the choice is very well described. The authors show multiple models under comparison, thus the setting of experiment is adequate. Ablation study and qualitative analysis are also a good contribution to the work.

Even though the idea of using entity typing for ED I snot novel, the approach presented in the paper is interesting and the authors evaluate and show improvement (SotA and to the base approach) on multiple datasets. Moreover, it is less computationally costly, as it does not use generative models, but achieves similar results.

**Reasons To Reject:**

This paper might be considered as an incremental study as it just adds another loss function to the main computation. However, duck typing along with polar box embeddings significantly improve the results, therefore, I find the amount of contribution sufficient for publication.

As a recommendation for quantitative analysis / error analysis, it would be interesting to look into relation boxes with mostly incorrect entities and try to understand what confused the model most?

**Reproducibility:**

4: Could mostly reproduce the results, but there may be some variation because of sample variance or minor variations in their interpretation of the protocol or method.

**Reviewer Confidence:**

4: Quite sure. I tried to check the important points carefully. It's unlikely, though conceivable, that I missed something that should affect my ratings.

---

> ### Author Rebuttal · Authors · 2023-08-28
>
> We thank the reviewer for the time spent on our paper and the overall positive feedback.
>
> We agree that our main contribution is an addition of a loss term to model the type of an entity, based on a method inspired by duck typing and relying on polar box embeddings. We believe this is a strength of our approach, as it considerably boosts downstream performance while being independent of the model used for the mention and entity encoder. Due to its simplicity, our approach can be, in principle, combined with a wide range of methods based on dense retrieval. We chose to build on the work of Wu et al. (2020) as it is computationally efficient and widely used in real-world applications (Plekhanov et al. 2023; Ayoola et al. 2022).
>
> We thank the reviewer for the question on the qualitative analysis. During our experiments, we noticed that boxes with a large fraction of incorrect entities often correspond to relations that, in Wikidata, are used to provide references to other knowledge bases (e.g., _“BabelNet ID”_). These relations do not carry type information, therefore it is difficult for the model to predict them based on the label and description of an entity.

---

### Meta-Review · Area_Chair_xH8A · 2023-09-18

**Recommendation:** 5

**Metareview:**

The reviewers found the paper well-written, easy to follow, and addressing an important problem. The reasons to reject are few and seem easy to address before camera ready. For example, publishing the source code could help readers grapple with the proposed method's complexity. During the author response period, the authors confirmed that they will publish the source code together with the paper.

---

### Decision · Program_Chairs · 2023-10-07

**Decision:**

Accept-Main

**Comment:**

The reviewers found the paper well-written, easy to follow, and addressing an important problem. The reasons to reject are few and seem easy to address before camera ready. For example, publishing the source code could help readers grapple with the proposed method's complexity. During the author response period, the authors confirmed that they will publish the source code together with the paper.